# Polylactide Nanocapsules Attenuate Adverse Cardiac Cellular Effects of Lyso-7, a Pan-PPAR Agonist/Anti-Inflammatory New Thiazolidinedione

**DOI:** 10.3390/pharmaceutics13091521

**Published:** 2021-09-20

**Authors:** Giani M. Garcia, Jérôme Roy, Ivan R. Pitta, Dulcinéia S. P. Abdalla, Andrea Grabe-Guimarães, Vanessa C. F. Mosqueira, Sylvain Richard

**Affiliations:** 1PhyMedExp, Inserm U1046, CNRS UMR 9214, Université de Montpellier, 34270 Montpellier, France; gianimg@gmail.com (G.M.G.); jerome.roy@inra.fr (J.R.); 2Department of Pharmacy, School of Pharmacy, Federal University of Ouro Preto, Ouro Preto 35400-000, Brazil; grabe@ufop.edu.br (A.G.-G.); mosqueira@ufop.edu.br (V.C.F.M.); 3Center of Health Sciences, Federal University of Pernambuco, Recife 50670-420, Brazil; irpitta@gmail.com; 4Department of Clinical and Toxicological Analyses, Faculty of Pharmaceutical Sciences, University of São Paulo, São Paulo 05508-000, Brazil; dspabdalla@gmail.com

**Keywords:** peroxisome proliferator-activated receptor-γ, cardiotoxicity, metabolic syndrome, atherosclerosis, contraction, calcium transient, ectopic diastolic Ca^2+^ events, polymeric nanoparticle

## Abstract

Lyso-7 is a novel synthetic thiazolidinedione, which is a receptor (pan) agonist of PPAR α,β/δ,γ with anti-inflammatory activity. We investigated the cardiotoxicity of free Lyso-7 in vitro (4.5–450 nM), and Lyso-7 loaded in polylactic acid nanocapsules (NC) in vivo (Lyso-7-NC, 1.6 mg/kg). In previous work, we characterized Lyso-7-NC. We administered intravenously Lyso-7, Lyso-7-NC, control, and blank-NC once a day for seven days in mice. We assessed cell contraction and intracellular Ca^2+^ transients on single mice cardiomyocytes enzymatically isolated. Lyso-7 reduced cell contraction and accelerated relaxation while lowering diastolic Ca^2+^ and reducing Ca^2+^ transient amplitude. Lyso-7 also promoted abnormal ectopic diastolic Ca^2+^ events, which isoproterenol dramatically enhanced. Incorporation of Lyso-7 in NC attenuated drug effects on cell contraction and prevented its impact on relaxation, diastolic Ca^2+^, Ca^2+^ transient amplitude, Ca^2+^ transient decay kinetics, and promotion of diastolic Ca^2+^ events. Acute effects of Lyso-7 on cardiomyocytes in vitro at high concentrations (450 nM) were globally similar to those observed after repeated administration in vivo. In conclusion, we show evidence for off-target effects of Lyso-7, seen during acute exposure of cardiomyocytes to high concentrations and after repeated treatment in mice. Nano-encapsulation of Lyso-7 in polymeric NC attenuated the unwanted effects, particularly ectopic Ca^2+^ events known to support life-threatening arrhythmias favored by stress or exercise.

## 1. Introduction

Peroxisome proliferator-activated receptors (PPARs) are members of the nuclear receptor family and function as ligand inducible transcription factors to regulate genes involved in lipid and glucose metabolism [1,2]. Thiazolidinediones (TZDs) are PPARγ agonists widely used as antihyperglycemic agents to improve the liver, muscles, and adipose tissue insulin sensitivity and treat type 2 diabetes mellitus [3]. The TZDs have additional anti-inflammatory and atheroprotective effects, yet the activation of PPARα may also be involved [4]. Despite reported favorable effects on cardiovascular risk factors, ischemic heart diseases and cardiovascular events may increase in diabetic patients [5,6]. TZDs also increase the risk of developing heart failure (HF) [7,8,9]. Treatment with TZD, such as rosiglitazone or pioglitazone, increase stroke risk in patients [10].

Lyso-7 is a new indole-thiazolidine compound (Figure 1A) recently synthesized [11]. It is a promising hybrid molecule acting as a pan partial agonist of PPARγ, PPARα, and PPARβ/δ but also acting as a COX-2 inhibitor [11,12]. Lyso-7 has a potent in vivo beneficial effect on inflammation and microcirculatory damage and the development of atherosclerotic lesions [13]. Lyso-7 exhibits anti-inflammatory effects via inhibition of expression of adhesion proteins, abolishing the adhesion of neutrophils to endothelial cells, and effects on blocking the enhancement of intracellular Ca^2+^ levels in neutrophils [14].

Polymeric nanocapsules (NC) (Figure 1A) are oily core nanostructures surrounded by a polymeric wall [15,16]). They can encapsulate different lipophilic molecules at high payloads and are biodegradable. They have demonstrated outstanding potential to reduce the cardiotoxicity of several drug candidates such as artemether, halofantrine, and lychnopholide, as reported by our group [17,18,19,20]. These properties may reflect NC’s ability to control drug release in the blood, reducing the free fraction producing potentially toxic adverse effects. Lyso-7 is a lipophilic compound (*c*log·*P* = 5.6). After encapsulation in NC, its physicochemical properties, release kinetics, and intravenous pharmacokinetics have been characterized [21]. NC significantly improves the Lyso-7 biopharmaceutical profile. Lyso-7 NC is suitable for oral or intravenous administration due to its nanometrical size dispersion. After intravenous administration in mice, compared with free Lyso-7 administered in solution, NC retained Lyso-7 in the plasma compartment, increasing 14-fold the plasma concentration and 3-fold the heart concentration. Upon intravenous administration, the accumulation of Lyso-7 in the heart was unrelated to blood flow. No apparent cardiotoxicity of Lyso-7 was observed in vivo when the NC formulation was administered intravenously at a dose of 1.6 mg/kg [21].

Due to the TZD class of drugs’ clinical restrictions, our study now aimed to assess the cardiac cellular effects of intravenous repeated administrations of Lyso-7 (free form) and Lyso-7 encapsulated in polylactic acid NC in an experimental rodent model (mice). We investigated the effect of Lyso-7 on the contraction and Ca^2+^ handling in single left ventricular (LV) myocytes in vitro and after treatment in vivo. Primary objectives were to: (i) detect off-target effects potentially leading to unexpected harmful side effects, and (ii) determine if the use of NC provides cardiac protection.

## 2. Materials and Methods

### 2.1. Materials

Lyso-7, [(5Z)-5-[5-bromo-1H-indol-3-yl)methylene]-3-(4-chlorobenzyl)-thiazolidine-2,4-dione] (CAS Number 1505484-42-3) (CAS Registry Number 1505484-42-3), Mw 447.73 g/mol, was synthesized, purified and characterized by Prof Ivan da Rocha Pitta as reported [11]. PLA polymer, a poly-rac-lactide Mn 75,000–120,000 g/mol, poloxamer 188, polysorbate 80, polyethylene glycol 300 (PEG 300), glucose, ethanol and methanol (HPLC grade) were purchased from Sigma-Aldrich (Cotia, Brazil). Symplicity^®^ System (Millipore, Bedford, MA, USA) produced Milli-Q water to prepare all solutions throughout the experiments. Soy lecithin with 75% of phosphatidylcholine (Epikuron^®^ 170) was a generous gift from Lucas Meyer (Champlan, France). Sasol Olefins & Surfactants GmbH (Hamburg, Germany) provided Miglyol 810 N. Tedia (Rio de Janeiro, Brazil) provided ethyl acetate, acetone, N,N-dimethylacetamide (DMA), acetone and dimethylsulfoxide (DMSO) (analytical grade).

### 2.2. Preparation of Lyso-7 Solution and Lyso-7-Loaded Nanocapsules

Lyso-7 was dissolved in absolute ethanol further diluted in buffers used in each experiment with ethanol concentration no higher than 2% *v*/*v*. We used this solution without Lyso-7 as a control in the acute experiments performed in vitro with isolated cardiomyocytes. We prepared the free Lyso-7 solution for intravenous (iv) treatment of mice as follows: we dissolved Lyso-7 in 20 μL of DMSO, 20 μL of polysobate 80, 380 μL of DMA, and 580 μL of PEG 300. This solution was mixed for 10 min and then appropriately diluted in isotonic glucose to obtain 0.5 mg/mL of free Lyso-7 solution used in vivo. We used the same solution without Lyso-7 as a control in vivo.

We used the nanoprecipitation method to make monodispersed nanocapsules loading Lyso-7 (Lyso-7-NC) or without Lyso-7 (blank-NC) as previously described [21]. We prepared Lyso-7-NC with 0.6% *wt*/*v* of PLA, 0.75% *v/v* of Miglyol 810N), and 0.75% *wt*/*v* of soybean lecithin dissolved in 10 mL of acetone. We added this organic solution into 0.75% *wt*/*v* of Poloxamer 188 solution (20 mL) via a syringe. We maintained dispersion under magnetic stirring for 10 min, and then all organic solvents and part of the water were evaporated under reduced pressure (Heidolph Rotary Evaporator, Schwabach, Germany) to render 5 mL of the aqueous suspension of NC.

The mean hydrodynamic diameters were 296 ± 4.2 nm and 265 ± 1.7 nm, respectively, with dispersion indexes less than 0.3 indicating the monodispersed population of particles in size. NC encapsulated Lyso-7 with high efficiency (83%) at a 0.5 mg/mL Lyso-7 concentration, attributed to Lyso-7 high lipophilicity. We have already characterized these NC in detail [21]. We used blank-NC as controls vs. Lyso-7-NC. All the formulations of Lyso-7, the 0.5 mg/mL iv solution (Lyso-7) and the 0.5 mg/mL Lyso-7 NC (Lyso-7-NC), as well as blank-NC, were passed through a 0.8 μm sterile filter before appropriate dilution in isotonic glucose to allow intravenous injection in mice of the doses established.

### 2.3. Animals and Cardiomyocytes

We used 7-week-old male C57BL/6J mice (Janvier Labs, Le Genest-Saint-Isle, France) bred and housed (4 mice/cage) under pathogen-free conditions (22 ± 2 °C; 12-h day/12-h night cycle) with standard rodent chow diet, *ad libitum* access to water, and behavioral needs (wood bedding material, cardboard tunnel, nesting material, and wooden gnawing sticks). For repeated-dose intravenous administration (via tail vein), mice were blindly randomized to the four different groups: vehicle (intravenous solution diluted in isotonic glucose), Lyso-7 (1.6 mg/kg/day), blank-NC (equivalent dose of NC excipients related to 1.6 mg/kg/day), and Lyso-7-NC (1.6 mg/kg/day). We administered mice vehicle and blank-NC as control groups for Lyso-7 and Lyso-7-NC, respectively, for comparisons. Repeated doses of both Lyso-7 and Lyso-7-NC for seven days produced no death in mice. We isolated myocytes freshly from mice of all experimental groups after the seven days of repeated administrations, as shown in Figure 1B.

We assessed the acute effects of Lyso-7 in vitro following the application of 4.5 nM, 45 nM, and 450 nM of Lyso-7 for 15 min on single cardiomyocytes isolated from untreated animals. The hearts were excised after euthanasia of mice by cervical dislocation to ensure rapid death without injecting any substance interfering with cellular physiology. Whole hearts were submitted to the liberase action using a Langendorf perfusion system to obtain individual LV myocytes as described [18,20,22,23,24]. We used only quiescent cardiomyocytes with precise edges. Otherwise, we investigated cells randomly. We carried out the experiments 1 to 4 h after cells enzymatic isolation. Figure 1B shows the general schema of animal experiments and protocols.

### 2.4. Measurements of Contraction and Intracellular Ca^2+^ in Single Cardiomyocytes

We field-stimulated myocytes with 1-ms current pulses delivered at 1 Hz to assess cell sarcomere length (SL) shortening, an index of contraction, and intracellular Ca^2+^ transients recorded simultaneously using an IonOptix system (IonOptix LCC. Milton, MA, USA) with a Zeiss microscope (Carl Zeiss GmbH, Oberkochen, Germany) (40× oil-immersion objective, 0.36 μm/pixel) as described [18,20,22,23,24]. Briefly, cells were bathed in a solution containing (in mM) 117 NaCl, 5.7 KCl, 11 glucose, 1.7 MgCl_2_, 1.8 CaCl_2_, 4.4 NaHCO_3_, 1.5 KH_2_PO_4_, and 21 HEPES (pH 7.4); all chemicals bought from Sigma-Aldrich, Saint-Quentin-Fallavier, France. We loaded cells with the dual-emission ratiometric Ca^2+^ indicator Indo-1AM (2 µM, Invitrogen, Grand Island, NY, USA) to monitor cytosolic Ca^2+^ determined by the ratio of 405 nm/480 nm fluorescence (a.u.: arbitrary units) [18,20,22,23,24]. We paced cells for 30 s periods followed by a 30-s rest period to investigate spontaneous Ca^2+^ diastolic events and contractions. In some experiments, we mimicked stress conditions by exposing cells to the β-adrenergic agonist isoproterenol (ISO, 10 nM) for 5 min before the experiments. Data were analyzed using Ionwizard^®^ Software (version 7.4, IonOptix®, Westwood, LA, USA).

### 2.5. Sparks Confocal

We visualized Ca^2+^ sparks in quiescent myocytes incubated with the Ca^2+^ indicator Fluo-4AM (4 μM) (Molecular Probes Inc., Eugene, OR, USA), visualized by confocal imaging with a Zeiss LSM510 microscope (Carl Zeiss Inc., Oberkochen, Germany) equipped with 63×/1.2 N.A. water immersion objective, at 25 °C in line-scan mode (1.5 ms/line, 512 pixels × 3000 lines) as described [18,22,23]. The dye was excited at 488 nm, and the fluorescence emission was collected through a 505-nm long-pass filter. Myocytes were field-stimulated at 1 Hz with 1-ms current pulses delivered via two platinum electrodes, one on each side of the perfusion chamber. During the rest period that followed stimulation, myocytes were repetitively scanned along the entire length of the cell at 1.5-ms intervals, for a maximum of 6 s. We reduced the laser intensity to 5% maximum to decrease cell damage and dye bleaching. Ca^2+^ sparks were determined by averaging the intensity of each sequential scan line and plotting the mean intensity as a function of time. We used the *SparkMaster* plugin for *ImageJ* software (Wayne Rasband, National Institutes of Health, Bethesda, MD, USA) to detect and analyze Ca^2+^ sparks.

### 2.6. Statistical Analysis

We performed statistical analyses using GraphPad Prism^®^ Software (Version 6.0, GraphPad Software Inc., San Diego, CA, USA). We expressed all data as mean ± SEM. We used D’Agostino and Pearson omnibus and Shapiro–Wilk normality tests, then a One-way ANOVA test for multiple comparisons. We performed post hoc tests where F was significant (*p* < 0.05) and no variance inhomogeneity. We used a Tukey post-test to compare all pairs of columns. We used a *t*-test for Ca^2+^ waves and Ca^2+^ sparks. *p* < 0.05 indicated a significant difference. N indicates the number of mice, and n indicates the number of independent cardiomyocytes isolated from these mice.

## 3. Results

### 3.1. Effect of In Vivo Repeated Dose Administration of Lyso-7 on Single Myocytes

#### 3.1.1. Differential Effects of Lyso-7 and Lyso-7-NC on Cellular Contraction and Ca^2+^

We investigated the effects of the repeated administration of Lyso-7 (free form) on contraction and Ca^2+^ transient evoked by electrical stimulation of LV myocytes at 1 Hz. No mice died under treatment with the formulations. First, it is worth noting that blank-NC did not affect any parameter relative to the control group (Figure 2, white bars). In contrast, Lyso-7 decreased cell contraction (−64%, SL shortening) and accelerated relaxation (25%) during systolic activity while not affecting resting SL (Figure 2A–D). In parallel, Lyso-7 lowered diastolic Ca^2+^ (−5%), reduced the Ca^2+^ transient amplitude (−35%) and accelerated the Ca^2+^ transient decay kinetics (16%) (Figure 2E–H). However, when Lyso-7 was incorporated in NC, its effect on contraction was less pronounced, by a factor of two (−33%, Lyso-7-NC vs. NC; Figure 2C, ! *p* < 0.05), and the impact of Lyso-7 on cell relaxation was prevented (Figure 2D) as well as its effect on the diastolic Ca^2+^, Ca^2+^ transient amplitude and Ca^2+^ transient decay kinetics (Figure 2F–H, ! *p* < 0.05).

We next evaluated the effects of Lyso-7 under the challenge of a maximally active concentration of the β-adrenergic receptor agonist isoproterenol (ISO,10 nM) to mimic the effects of physical exercise and stress [25]. We exposed myocytes to ISO for 5 min before recordings. As expected, in all experimental groups (compare averaged data in Figure 3 with corresponding values in Figure 2), acute exposure of cells to ISO enhanced cell contraction, accelerated relaxation, and, in parallel, increased Ca^2+^ transient amplitude and accelerated Ca^2+^ transient decay kinetics in line with enhanced reuptake of Ca^2+^ in the sarcoplasmic reticulum (SR) via the SERCA2a pump [26]. For example, in the control group, ISO increased cell contraction by 43%, accelerated relaxation by 29%, increased Ca^2+^ transient amplitude by 81%, and accelerated Ca^2+^ transient decay kinetics by 60%. Overall, these effects were consistent with those reported in the literature [25]. However, ISO had a two-fold more positive inotropic effect on contraction, compensating for the decreasing impact of Lyso-7 as found in basal conditions while also enhancing the lusitropic impact (Figure 3B,C). In the same line, free Lyso-7 still lowered diastolic Ca^2+^ and accelerated the Ca^2+^ transient decay kinetics (Figure 3D,F), similarly to the effects observed in the absence of ISO (Figure 2F,H) except for the Ca^2+^ transient where the decreasing impact of ISO was partly compensated (compare Figure 2G with Figure 3E). The different effects of Lyso-7 in the presence of ISO were absent or attenuated by nano-encapsulation (Lyso-7-NC formulation). In basal conditions and with ISO, these data showed that Lyso-7 impacts cellular contraction and intracellular Ca^2+^, but its effects could be mostly prevented or significantly attenuated (e.g., contraction) by encapsulation in NC.

#### 3.1.2. Differential Effects of Lyso-7 and Lyso-7-NC on Ectopic Ca^2+^ Events at Rest

The repeated administration of Lyso-7 promoted ectopic diastolic Ca^2+^ events, not observed in the control and NC groups, during resting periods following a train of stimulations (Figure 4A,C). When we incorporated Lyso-7 in NC (Lyso-7-NC group), we detected no significant abnormal activity. We also observed no irregular Ca^2+^ transient during pacing in any of the four groups. We next tested the effect of an ISO challenge on the promotion of spontaneous Ca^2+^ events. First of all, ISO (10 nM), as such, favored their occurrence (Figure 4B,D) in line with reports by others [27,28,29]. Nevertheless, Lyso-7 severely enhanced the number of cells exhibiting ectopic Ca^2+^ events (80% vs. 10% in the absence of ISO), which was abolished in the Lyso-7-NC group (Figure 4D). Therefore, Lyso-7 induced aberrant diastolic Ca^2+^ events, severely increased by ISO but controlled by NC.

### 3.2. Acute Effects of Free Lyso-7 on Single Cardiomyocytes In Vitro

#### 3.2.1. Lyso-7 Impairs Contraction and Ca^2+^ Transient

In separate experiments, we acutely exposed cardiomyocytes from untreated animals for 15 min to Lyso-7 at 4.5 nM, 45 nM, and 450 nM, respectively (Figure 5). The highest concentration corresponded to 200 µg/mL, i.e., a concentration almost one hundred times higher than the blood concentration as determined in studies performed in mice in vivo (1.86 µg/mL; [21]). Free Lyso-7 increased contraction, but only at 450 nM (Figure 5A,C). Lyso-7 also accelerated cellular relaxation dose-dependently (+30% at 450 nM), an effect detected at 45 nM (17%) (Figure 5D). Lyso-7 did not affect resting SL (Figure 5B). In parallel, Lyso-7 at 450 nM lowered diastolic Ca^2+^ (−29%) (Figure 5E,F) and reduced the amplitude of the Ca^2+^ transient (−45%) (Figure 5E,G). In contrast, Lyso-7 did not affect the decay of the Ca^2+^ transient (Figure 5H).

Acute exposure to ISO increased SL shortening (i.e., augmented contraction) approximately twice, except at 450 nM (+25%) due to the intrinsic positive effect of Lyso-7 at this concentration, and enhanced relaxation (+30%) in control myocytes (compare values in Figure 6C,D and Figure 5C,D). In parallel, ISO increased the amplitude of the Ca^2+^ transient (+80%), the diastolic Ca^2+^ (+12%), and accelerated the Ca^2+^ transient decay (+40%) (compare values in Figure 6F–H and Figure 5F–H). Under these experimental conditions (presence of ISO), we observed no effect of Lyso-7 on cell contraction and relaxation, even at 450 nM (Figure 6A,C). In particular, there was no additive inotropic effect. Similarly, Lyso-7 did not affect the Ca^2+^ transient amplitude, diastolic Ca^2+^, and Ca^2+^ transient decay in the presence of ISO at 4.5 nM and 45 nM. However, at 450 nM, Lyso-7 still reduced both the resting Ca^2+^ (−37% vs. control) and the Ca^2+^ transient amplitude (−47%; Figure 6F,G) and reminded the effects of Lyso-7 in basal conditions (absence of ISO) (Figure 5F,G). In contrast, Lyso-7 did not affect the decay of the Ca^2+^ transient (Figure 6H), as seen in the absence of ISO (Figure 5H).

#### 3.2.2. Lyso-7 Promotes Abnormal Ca^2+^ Events at Rest

We detected no abnormal activity, either during pacing or rest, in control cells for low concentrations of Lyso-7. However, Lyso-7 at 450 nM promoted diastolic ectopic Ca^2+^ events in nearly 20% of cells (Figure 7A) though they were small and infrequent. The presence of abnormal macroscopic Ca^2+^ events at rest often results from a ryanodine receptor 2 (RyR2) leakage, generating microscopic Ca^2+^ events called Ca^2+^ sparks [27]. Consistent with this possibility, Lyso-7 at 450 nM increased the onset of Ca^2+^ sparks (+125%; Figure 7C,D). We next tested the effect of Lyso-7 in cells subjected to an ISO challenge. Once again, ISO (10 nM), as such, promoted the occurrence of ectopic Ca^2+^ waves (in 30% of cells, Figure 7A) and Ca^2+^ sparks (approximately two-fold, Figure 7C,D). Lyso-7 had no further impact at 4.5 nM and 45 nM in the presence of ISO, but it was worth noting that Lyso-7 severely enhanced the number of cells exhibiting ectopic Ca^2+^ events (80% vs. 10% in the absence of ISO). In the presence of ISO, all cells exhibited spontaneous diastolic Ca^2+^ events at 450 nM (Figure 7A). Lyso-7 also further increased Ca^2+^ sparks frequency (+28%; Figure 7C,D). Therefore, Lyso-7 promoted unwanted spontaneous Ca^2+^ events, both microscopically and macroscopically, particularly during the β-adrenergic challenge of ISO in line with this pathway’s additive or synergic effect.

## 4. Discussion

The main result of our study is that the nano-encapsulation of Lyso-7 in PLA-NC protects cardiac cells from the deleterious off-target effects of this drug. Indeed, Lyso-7 free impacted several parameters of the cellular excitation-contraction coupling and, in particular, promoted abnormal spontaneous firing of undesirable ectopic diastolic Ca^2+^ events known to increase the arrhythmogenic risk in vivo. Nano-encapsulation of Lyso-7 abolished or significantly attenuated the unwanted effects.

A significant result of our study was evidence for some off-target effects of Lyso-7, seen during acute exposure of cardiomyocytes to high concentrations or after repeated treatment in vivo. Indeed, most of the adverse effects, namely on contraction, intracellular Ca^2+^ handling, and pro-arrhythmogenic Ca^2+^ events, were retrieved after repeated intravenous administration of Lyso-7 at 1.6 mg/kg/day in mice for seven days. Of note, a counterintuitively divergent effect appeared on cell contraction (i.e., a decrease instead of an increase following acute exposure), suggesting different mechanisms or that cell physiology has been impacted during the repeated administration. Nevertheless, Lyso-7 had many similar effects and profoundly modified the Ca^2+^ homeostasis of the myocytes when administered acutely and directly on the cardiomyocytes or repeatedly to mice. Lyso-7 lowered the resting Ca^2+^, decreased the amplitude of the Ca^2+^ transient, and accelerated its decay kinetics. Finally, Lyso-7 promoted abnormal firing of Ca^2+^ events during diastolic periods in both experimental conditions.

Ca^2+^ sparks reflect discrete, elementary Ca^2+^ signaling events arising from abnormally leaky RyR2 channels [27,28]. They participate critically in initiating and propagating pro-arrhythmogenic spontaneous Ca^2+^ waves. From a mechanistic point of view, the processes involved here are unclear. RyR2 leakage may participate in depleting SR Ca^2+^ content, therefore explaining the decrease of Ca^2+^ transient amplitude. However, since this depletion should increase intracellular Ca^2+^, it was counterintuitive to find that Lyso-7 decreases diastolic Ca^2+^. The Ca^2+^ levels are controlled mainly by the balance between uptake of Ca^2+^ via SERCA2a and Ca^2+^ release of the RyR2 receptors. In case of imbalance, abnormal diastolic Ca^2+^ discharge activates the electrogenic Na^+^/Ca^2+^ exchanger (NCX), producing delayed afterdepolarization and potentially ectopic Ca^2+^ transients [26,27,30]. Therefore, Lyso-7 may enhance the Na^+^/Ca^2+^ exchanger (NCX) activity, essential for removing intracellular Ca^2+^ during relaxation [31], to account for the reduced diastolic Ca^2+^.

The β-adrenergic challenge did not change much the overall effect of Lyso-7 on the basic Ca^2+^ handling parameters, but it dramatically enhanced its impact on the firing of macroscopic Ca^2+^ events as well as Ca^2+^ sparks at rest. Low RyR2 leakage is insufficient *per se* to produce arrhythmogenic diastolic Ca^2+^ release, particularly with enhanced NCX-mediated Ca^2+^ extrusion, because of the depletion in the SR Ca^2+^ content [32]. However, under β-adrenergic stimulation, increased SR-Ca^2+^ content, consistent with Ca^2+^ transient decay acceleration found here, is known to enhance RyR2 open probability and the associated arrhythmia risks [32] in line with our findings on Ca^2+^ waves and Ca^2+^ sparks (Figure 4 and Figure 7). Overall, this result points out a genuine cumulative pro-arrhythmogenic risk of Lyso-7 under stress or physical exercise.

The effects of Lyso-7 were complex, probably involving several mechanisms. In contrast with its impact on Ca^2+^ handling, Lyso-7 had divergent acute and long-term effects on cell contraction (increase vs. decrease, respectively). The amount of Ca^2+^ released depends on the amount of Ca^2+^ stored in the SR [33]. The positive inotropic effect of Lyso-7 on contraction was uncorrelated with the concomitant reduction in the Ca^2+^ transient amplitude, which was paradoxical. Since this latter parameter is a crucial determinant of contraction, another mechanism must be responsible for the positive inotropy. An increase in the sensitivity of the contractile proteins to Ca^2+^ is a likely candidate. We did not investigate the underlying mechanism because that was not the main objective of our study. ISO could not prevent, or compensate for, the impact of Lyso-7 on Ca^2+^ handling parameters such as the decrease in diastolic Ca^2+^ and Ca^2+^ transient amplitude. Overall, our data suggest that Lyso-7 has multiple molecular off-targets, direct targeting of contractile protein activity being a likely possibility in addition to Ca^2+^ handling. Moreover, the divergent effect on contraction, particularly the decrease resulting from repeated exposure of cells to Lyso-7 in vivo, may reflect a long-term impact of the molecule on this parameter. This possibility warrants further thorough investigation on the effect of more prolonged, repeated administration to investigate whether morphological and functional changes occur during chronic exposure at the whole organ level and whether NC ensures long-term cardioprotection.

The most important result of our study was the safety provided by the administration of Lyso-7 in the NC dosage form for seven days. Indeed, Lyso-7 in NC was devoid of cardiotoxicity for most parameters investigated at the cellular level. Lyso-7-NC had only little effect on Ca^2+^ handling, spontaneous Ca^2+^ events, and contraction of cardiomyocytes. The overall result was somewhat unexpected because the administration of Lyso-7 encapsulated in polymeric NC increases Lyso-7 body exposure by 14-fold and heart concentration by 3.5-fold, respectively, according to the pharmacokinetics and biodistribution studies [21]. The lipophilic character of free Lyso-7 is also likely to direct the molecule preferentially to the cell membrane, contributing to the increase in the whole heart. Cell membranes would then act as a storage site, a reservoir, gradually releasing the active principle towards its targets over time. The prolonged release of Lyso-7 from the NC in vivo may significantly reduce the fraction of the free molecule and minimize the interaction with the cardiomyocytes and the off-target effects. This is consistent with the absence of the impact of free Lyso-7 when applied at low concentrations (<45 nM). In vitro, the time required to release 50% Lyso-7 from NC is 420 min compared to 30 min dissolution of free Lyso-7, indicating that NC is a sustained release device for this molecule [21]. From our previous pharmacokinetic data, the intravenous dose of 1.6 mg/kg is large enough to induce the effects on COX-1, COX-2, PPAR-α, PPAR-*β/δ* or PPAR-*γ* [12,13].

## 5. Conclusions

The association of Lyso-7 with PLA-NC is promising to provide a potential therapeutic option devoid of significant adverse cardiac effects in line with previous findings from our group showing that NC can attenuate the toxicity of artemether and lychnopholide both at the heart and cardiomyocytes levels [18,19,20]. The NCs were efficient to prevent aberrant Ca^2+^ events known to support life-threatening Ca^2+^-dependent ventricular arrhythmias potentially favored during stress or exercise. The interest of this concept can extend to all TZD to treat type 2 diabetes because of risks of increased HF and mortality incidence [34] and all the potential uses of Lyso-7. Although more investigation is warranted, our data may pave the way for NC in future applications.

## Figures and Tables

**Figure 1 pharmaceutics-13-01521-f001:**
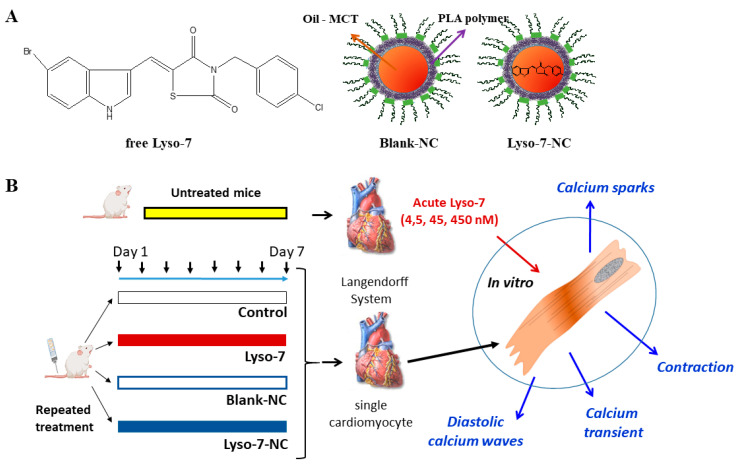
Chemical structure of Lyso-7 and schematic representation of polymeric nanocapsules (**A**). Presentation of the experimental protocol used to treat mice and isolate the cardiomyocytes to test cardiotoxicity of the formulations (**B**) directly in vitro (acute effects) or after in vivo treatment followed by cardiomyocyte isolation.

**Figure 2 pharmaceutics-13-01521-f002:**
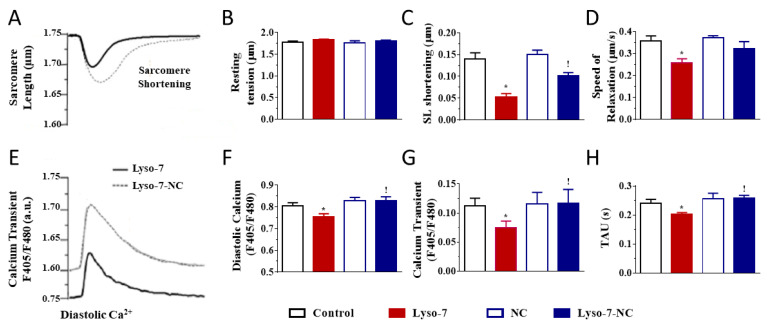
Effect of in vivo administration of Lyso-7 and Lyso-7-NC (1.6 mg/kg, once a day for seven days) on contraction (**A**–**D**) and Ca^2+^ transient (**E**–**H**) in freshly isolated cardiomyocytes. Typical recordings of the effects of Lyso-7 (free form) and Lyso-7-NC on sarcomere length (SL) shortening (**A**) and Ca^2+^ transient (**E**), both under field stimulation at 1 Hz. (**B**–**D**): averaged data of resting SL, SL shortening and relaxation, respectively, and (**F**–**H**): averaged data of diastolic Ca^2+^, Ca^2+^ transient and decay of Ca^2+^ transient (Tau), respectively, all for the four experimental groups (Control, Lyso-7, NC, and Lyso-7-NC). ANOVA followed by Tukey post-test (*p* < 0.05). * Lyso-7 vs. Control; ! Lyso-7-NC vs. Lyso-7. *n* = 4; *n* = 12–18.

**Figure 3 pharmaceutics-13-01521-f003:**
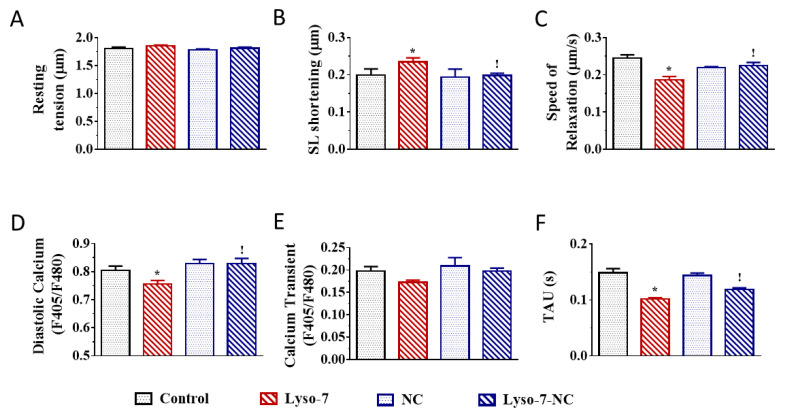
Effect of in vivo administration of Lyso-7 and Lyso-7-NC (1.6 mg/kg, once a day for seven days) on contraction (**A**–**D**) and Ca^2+^ transient (**E**–**H**) of freshly isolated cardiomyocytes exposed in vitro to Isoproterenol (ISO, 10 nM). (**A**–**C**): averaged data of resting sarcomere length (SL), SL shortening and relaxation, respectively, and (**D**–**F**): averaged data of diastolic Ca^2+^, Ca^2+^ transient and decay of Ca^2+^ transient (Tau), respectively, all for the four experimental groups (Control, Lyso-7, NC, and Lyso-7-NC). ANOVA followed by Tukey post-test (*p* < 0.05). * Lyso-7 vs. Control; ! Lyso-7-NC vs. Lyso-7. *n =* 4; *n* = 12–18.

**Figure 4 pharmaceutics-13-01521-f004:**
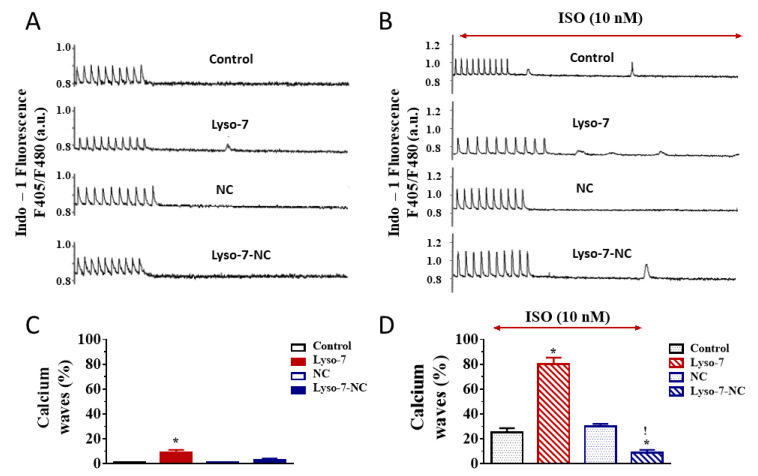
Effect of in vivo administration of Lyso-7 and Lyso-7-NC (1.6 mg/kg, once a day for seven days) on cardiomyocytes’ ectopic diastolic Ca^2+^ waves. (**A**) Representative recordings of Ca^2+^ waves during resting periods after a train of field stimulation at 1 Hz in cardiomyocytes from the four experimental groups (Control, Lyso-7, NC, and Lyso-7-NC); (**B**) Representative recordings of Ca^2+^ waves during resting periods after a train of field stimulation at 1 Hz in cardiomyocytes from the four the same experimental groups treated in vitro with Isoproterenol (ISO, 10 nM); (**C**,**D**) percentage of cardiomyocytes developing at least one spontaneous Ca^2+^ wave in absence and presence of ISO (10 nM) in vitro, respectively, for the four experimental groups (Control, Lyso-7, NC, and Lyso-7-NC). We used a *t*-test. *p* < 0.05,* Lyso-7 vs. control; ! Lyso-7-NC vs. Lyso-7. a.u.: arbitrary units. *n* = 4; *n =* 12–18.

**Figure 5 pharmaceutics-13-01521-f005:**
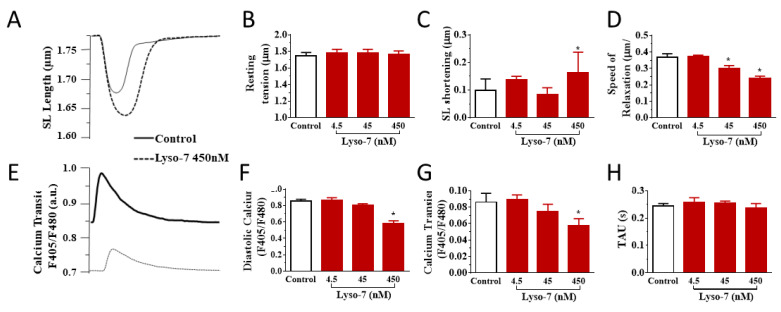
Acute effect of Lyso-7 on contraction (**A**–**D**) and Ca^2+^ transient (**E**–**H**) in cardiomyocytes of untreated mice. Typical recordings of Lyso-7 (450 nM) effects on sarcomere length (SL) shortening (**A**) and Ca^2+^ transient (**E**), both under field stimulation at 1 Hz. (**B**–**D**) averaged data of resting SL, SL shortening and relaxation, respectively, and (**F**–**H**): averaged data of diastolic Ca^2+^, Ca^2+^ transient and decay of Ca^2+^ transient (Tau), respectively, for Control, and increasing concentration of Lyso-7 (4.5, 45, and 450 nM). ANOVA followed by Tukey post-test (*p* < 0.05). * Lyso-7 vs. Control. Control refers to the absence of Lyso-7. *n =* 12–18.

**Figure 6 pharmaceutics-13-01521-f006:**
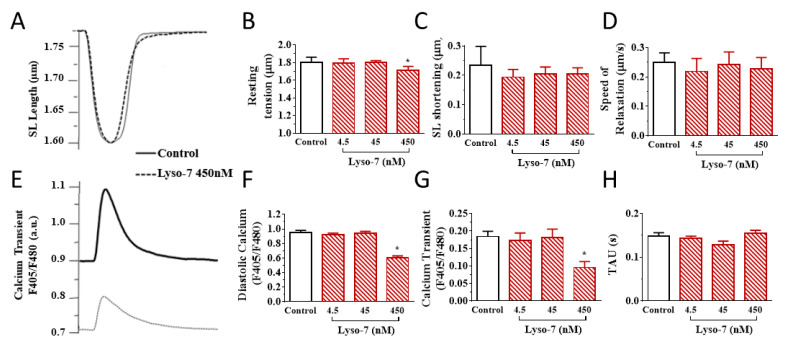
Acute in vitro effect of Lyso-7 on contraction (**A**–**D**) and Ca^2+^ transient (**E**–**H**) of cardiomyocytes under IsoproTable 10. nM). Typical recordings of Lyso-7 (450 nM) effects on (**A**) sarcomere length (SL) shortening and (**E**) Ca^2+^ transient, both under field stimulation at 1 Hz. (**B**–**D**) averaged data of resting SL, SL shortening and relaxation, respectively, and (**F**–**H**) averaged data of diastolic Ca^2+^, Ca^2+^ transient and decay of Ca^2+^ transient (Tau), respectively, for Control, and increasing concentration of Lyso-7 (4.5, 45, and 450 nM). ANOVA followed by Tukey post-test (*p* < 0.05). * Lyso-7 vs. Control. Control refers to the absence of Lyso-7. *n =* 12–18.

**Figure 7 pharmaceutics-13-01521-f007:**
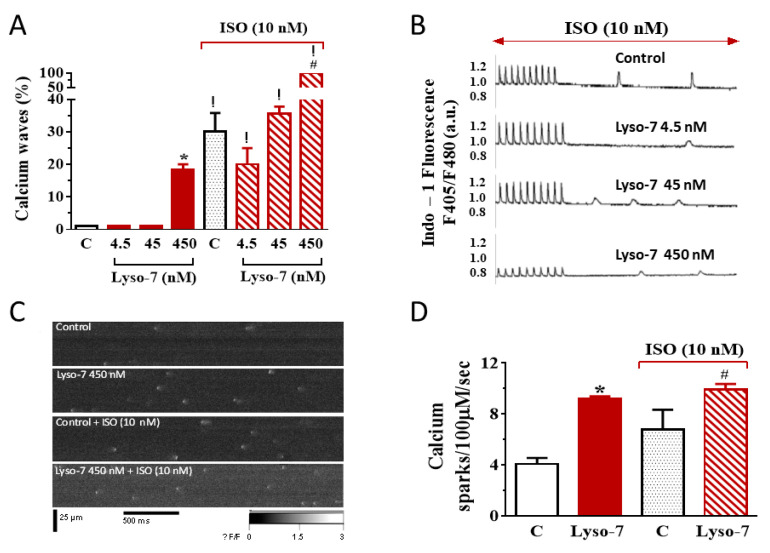
Acute in vitro effect of Lyso-7 on cardiomyocytes’ abnormal spontaneous diastolic Ca^2+^ events under IsoproTable 10. nM) challenge. (**A**) percentage of cardiomyocytes developing at least one spontaneous Ca^2+^ wave in the absence (left panel) and presence (right panel) of Isoproterenol (ISO, 10 nM) during resting periods after a train of field stimulation at 1 Hz in cardiomyocytes. (**B**) Representative recordings of Ca^2+^ waves under ISO. *n* = 12–18 cells. (**C**) Typical line-scan confocal images of Ca^2+^ sparks from Fluo-4-AM loaded cardiomyocytes; *n =* 7–10. (**D**) Mean of frequency of Ca^2+^ sparks measured in the different conditions indicated. C = Control refers to the absence of Lyso-7 (450 nM). We used a *t*-test. *p* < 0.05, * Lyso-7 vs. Control in the absence of ISO (left part of the panel); # Lyso-7 vs. Control in the presence of ISO (right part of the panel); ! with ISO vs. without ISO.

## Data Availability

The data presented in this study are available in the research article.

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
