# Peer review of "Polylactide Nanocapsules Attenuate Adverse Cardiac Cellular Effects of Lyso-7, a Pan-PPAR Agonist/Anti-Inflammatory New Thiazolidinedione"

_pharmaceutics, 2021, doi:10.3390/pharmaceutics13091521_

Round 1
Reviewer 1 Report
The authors studied the influence of the Lyso-7 and polylactide nanocapsules with Lyso-7 on cardiotoxicity. Part of the publication is comparing the effect of Lyso-7 and Lyso-7-NC. The other part is only about Lyso-7. Why you did not perform the in-vitro tests (3.2) also with the Lyso-7-NC? It will be nice to add it. This publication is well structured. Unfortunately, there are many grammatical mistakes (sentence order, wrongly used phrases) and long, non-consistent sentences. Therefore, some parts are hard to read and they do not make sense. Adding the characterization of nanoparticles is necessary. There is a citation in the text referring to the nanoparticles. However, in this citation (19), there is no mention of PLA nanoparticles. So, correct the citation and add nanocapsules characterization (DLS, AFM or other microscopy technique) including drug release profile! After major revisions and language corrections, I recommend this publication be published.
The particular remarks are listed below:
Abstract: Four followed sentences star the same. (21-24)
Introduction: Missing abbreviation Et-HCl (53)
What is so interesting about that? (60)
(62-63) This sentence – add at least among others.
(63-66) Long and incomprehensible sentence.
(94-95) We poured the solution with a syringe. For example: Added via syringe.
2.2 How many mice per group? According to what you chose dosing. The samples were administrated in-vivo via tail vein?
(125) Information concerning the survival of mice should be in results.
Did you check the weight of the mice? Were there any changes suggesting systemic toxicity?
2.5 (173) Missing verb.
Figure 2, Figure 5, Figure 6. The graphs are too small, 2Aa and 2Bb have poor quality.
Figure 4D: What is the explanation for so low Calcium wave for the Lyso-7-NC?
Figure 4C: Is it necessary to have 100 % on y bar, if highest obtained values are below 20%?
Figure 7Aa: Why there is no effect in the case of 45 nM of Lyso-7 (without ISO)? The trend should be the same as in case with ISO.
Figure 7Bb: Why there are no differences in the signals of Lyso-7 after adding the ISO?
(342) All of the unwanted effects were abolished? Figure 2Ac the effect was only suppressed.
(345-355) This part belongs to the introduction, not in the discussion.
Author Response
Please see PDF attached.

Reviewer 2 Report
The manuscript describes a polylactide nanocapsule delivery system that attenuates adverse cardiac cellular effects of Lyso-7. The authors incorporated Lyso-7 in polylactide nanoparticles which attenuated drug effects on cell contraction and prevented its impact on relaxation, diastolic Ca2+, Ca2+ transient amplitude, Ca2+ transient decay kinetics, and promotion of diastolic Ca2+ events. This concept is very interesting, and it can extend to other TZD to treat type 2 diabetes and the potential uses of Lyso-7. The experiments were thoughtfully designed and nicely carried out. The paper is enjoyable to read. Overall, this is a nice study and I recommend acceptance after considering the following points.
1. The molecular weight of PLA and the detailed methods of preparing the Lyso-7-NC should be provided in the Materials and Methods.
2. In Figure 2, the A,a and B,a are of low quality. Please improve the resolution of these figures.
3. Some parts in the Discussion section are repetitive and lengthy. Please reconstruct this part. For example, “Lyso-7 is a synthetic indole-thiazolidine molecule with multiple PPAR receptors targeting and promising hybrid actions to amplify and widen its therapeutic action while reducing its adverse effects. Lyso-7 acts as a receptor (pan) agonist of all three PPAR isoforms α,β/δ,γ, involving post-transcriptional mechanisms for the α and δ isoforms and transactivation or transrepression of target genes following PPARγ activation, and as a COX Inhibitor with anti-inflammatory activity.” has been mentioned multiple times in the main text.
4. Please make sure that abbreviations are introduced the first time a term appears and that they are used throughout. For example, when “ISO” appears in Abstract, both the full name and the abbreviation should be provided and then the abbreviation “ISO” should be used in the following part.
Author Response
See PDF attached

Reviewer 3 Report
The manuscript entitled “Polylactide nanocapsules attenuate adverse cardiac cellular effects of Lyso-7, a pan-PPAR agonist/anti-inflammatory new thiazolidinedione” is well organized and presents interesting and encouraging results. Please consider reviewing based on the suggestions below, in order to improve the article.
- Introduction:
The introduction part is quite brief.
- The originality of the research should be better highlighted in the manuscript.
- Please add the definition of nanopcapsules and what are their main advantages. I suggest some references that you can use: i) D. M. Rata, J.F Chailan, C. A. Peptu, M. Costuleanu, M. Popa, Chitosan: poly(N-vinylpyrrolidone-alt-itaconic anhydride) nanocapsules—a promising alternative for the lung cancer treatment, J Nanopart Res 17:316, (2015); ii) K. Z. Dellali, D. M. Rata, M. Popa, M. Djennad, A. Ouagued, D Gherghel. Antitumoral Drug: Loaded Hybrid Nanocapsules Based on Chitosan with Potential Effects in Breast Cancer Therapy. International Journal of Molecular Sciences, 21(16), 5659, (2020); iii) D.M. Raţă, M. Popa, J.F. Chailan, C.L. Zamfir, C.A. Peptu, Biomaterial properties evaluation of poly(vinyl acetate-alt-maleic anhydride)/chitosan nanocapsules. Journal of Nanoparticle Research, 16(8), (2014).
- Materials and Methods
On page 3, lines 105-106, The authors said that: “The mean hydrodynamic diameters were 296 ±4.2 nm and 265 ±1.7 nm, respectively, with a dispersion index less than 0.3 indicating the monodispersed population of particles in size.” Please add an analysis (DLS, SEM) to confirm the diameter of these nanocapsules.
Also, on page 3, line 107, the authors said that: “NC encapsulated Lyso-7 with a high encapsulation efficiency (83%) at a 0.5 mg/ml Lyso-7 concentration”. Please explain how you quantified the amount of Lyso-7 into nanocapsules.
Author Response
see PDF attached

Round 2
Reviewer 1 Report
Authors answered successfully most of my questions and tried to correct the paper according to suggestions. Therefore, I recommend to publish the paper in present form.